# Ambient Seismic Noise and Microseismicity Monitoring of a Prone-To-Fall Quartzite Tower (Ormea, NW Italy)

**Chiara Colombero** [1,*]**, Alberto Godio** [1] **and Denis Jongmans** [2]

1   Department of Environmental, Land and Infrastructure Engineering (DIATI), Politecnico di Torino, 10129 Torino, Italy; alberto.godio@polito.it
2   University Grenoble Alpes, University Savoie Mont Blanc, CNRS, IRD, IFSTTAR, ISTerre, 38000 Grenoble, France; denis.jongmans@univ-grenoble-alpes.fr
*   Correspondence: chiara.colombero@polito.it

**Abstract:** Remote sensing techniques are leading methodologies for landslide characterization and monitoring. However, they may be limited in highly vegetated areas and do not allow for continuously tracking the evolution to failure in an early warning perspective. Alternative or complementary methods should be designed for potentially unstable sites in these environments. The results of a six-month passive seismic monitoring experiment on a prone-to-fall quartzite tower are here presented. Ambient seismic noise and microseismicity analyses were carried out on the continuously recorded seismic traces to characterize site stability and monitor its possible irreversible and reversible modifications driven by meteorological factors, in comparison with displacement measured on site. No irreversible modifications in the measured seismic parameters (i.e., natural resonance frequencies of the tower, seismic velocity changes, rupture-related microseismic signals) were detected in the monitored period, and no permanent displacement was observed at the tower top. Results highlighted, however, a strong temperature control on these parameters and unusual preferential vibration directions with respect to the literature case studies on nearly 2D rock columns, likely due the tower geometric constraints, as confirmed by 3D numerical modeling. A clear correlation with the tower displacement rate was found in the results, supporting the suitability of passive seismic monitoring systems for site characterization and early waning purposes.

**Keywords:** passive seismic monitoring; landslides; ambient seismic noise; microseismicity; 3D numerical modeling; reversible modifications

## 1. Introduction

Continuous passive seismic monitoring has now reached a decade of applications on gravitational movements of all types and geometries [1]. Passive seismic monitoring systems usually involve a set of spatially distributed sensors deployed on, around or inside the potentially unstable compartments. The network continuously records the ambient seismic noise at the site. Spectral analysis and cross-correlation of the recorded noise are applied to extract resonance frequency variations and seismic velocity changes within the investigated volumes. Both seismic parameters can show reversible fluctuations driven by external modification in air temperature and precipitation [1] and irreversible drops in their values when failure is approached [2–5].

The spatial orientation of the resonance frequencies can also identify the potential direction of collapse and give further information about the constraints of the potentially unstable volume. For vertically elongated rock columns and prisms, separated from the stable rock mass by one or more near-vertical fractures, the first vibration mode is generally related to the bending of the unstable volume, perpendicular to the direction of the main fractures separating stable and unstable compartments [5–10]. The second vibration mode usually bends perpendicular to the first mode, i.e., parallel to the fracture orientation. More complex behaviors are detected at higher modes and on sites with 3D geometries [11]. As

for the reversible fluctuations of resonance frequency values, the vibration orientations and the spectral amplitudes can also vary with time as a result of the external forcing [12].

On the other hand, continuous passive seismic monitoring systems allows for the study of the microseismicity induced by the landslide movement or pre-failure seismic signals. Microfracturing processes within the unstable compartments release elastic waves that can be recorded by a set of spatially distributed sensors at the surface. These seismic events are extracted from ambient seismic noise recordings by means of detection algorithms. Their classification, source location and temporal trends can help to identify the most unstable compartments and track the evolution to failure [5,13–15].

In such a frame, both ambient seismic noise and microseismicity analyses are suitable tools for early-warning purposes and can be applied complementary or alternatively to remote sensing techniques for characterization and monitoring studies.

Here, we discuss the results of a six-month passive seismic monitoring experiment on a potentially unstable quartzite tower at the back of an active landslide. The tower is located along a rock cliff with large near-vertical discontinuities that generate isolated rock pinnacles and towers. For site characterization, a photogrammetric study was recently carried out in the area [16] to measure the main joint sets and identify the most unstable compartments in the inaccessible rock walls. The complex topography and dense vegetation highlighted the main limitations of terrestrial photogrammetry and laser scanning, suitable to acquire data only for limited portions of the cliff. The use of a remotely piloted aircraft system partially succeeded in overcoming these limitations, thanks to the acquisition of nadir and oblique data sets. However, the presence of vegetation at the top and on both sides of the tower prevented the reconstruction of its complete geometry and rear constraints.

A seismic monitoring network was deployed on the tower to further investigate and continuously monitor site stability. Continuous ambient seismic noise recordings were processed with spectral analysis and cross-correlation techniques to identify and track the resonance frequencies of the tower and possible seismic velocity changes related to modifications in its stability. Ground motion directions were also retrieved to understand the fracture control on tower vibrations, supported and verified by 3D numerical modeling. A large dataset of microseismic events was extracted from ambient seismic noise recordings. Event waveforms were classified on the basis of salient time and frequency domain features and related to different source mechanisms. The temporal modifications of both ambient seismic noise and microseismicity parameters were finally compared with on-site displacement measurements and meteorological parameters to gain insight into site stability and test the applicability of the methods for further early warning purposes.

## 2. Test Site and Monitoring Network

The study site is located on the western side of the Tanaro Valley, close to the town of Ormea (CN, NW Italy, Figure 1a). This forested valley side is characterized by widespread outcrops of metasedimentary, fine-grained quartzite belonging to Lower Triassic continental-to-transitional deposits of the Maritime Alps, which lay unconformably over Permian volcanosedimentary deposits [17]. The quartzite layering in near-horizontal, but the widespread presence of near-vertical fractures led to the generation of isolated pinnacles and towers (Figure 1b). In the last few years, a landslide occurred at the base of one of this quartzite towers, involving shallow slope deposits and a fractured rock mass (Figure 1b). The active landslide and the quartzite tower are separated by a fracture with wide opening (1 m approximately close to C1, Figure 2d).

Both the active landslide and the quartzite tower are continuously monitored by ARPA Piemonte (the regional agency for environment protection) and the Ormea Municipality by means of a set of extensometers located at the rear fracture of the active landslide and at the top of the quartzite tower (C1 to C3 in Figure 2). The increasing fracture opening and episodic material sliding at the base can potentially and progressively destabilize the tower

(approximately 3000 m$^3$) and lead to rockfalls, block toppling or complete tower collapse threating the riverbed, buildings and infrastructures downslope.

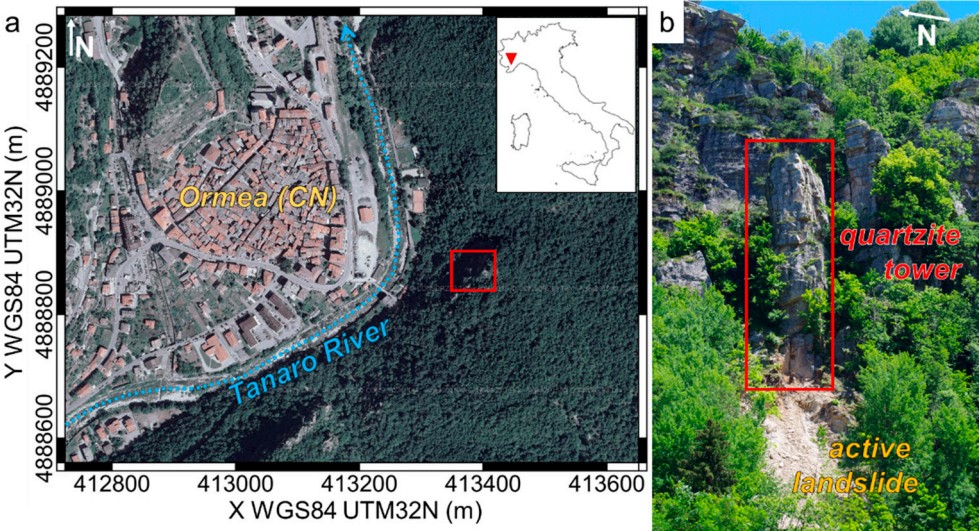

**Figure 1.** (**a**) Geographical location of the study site (Ormea, CN, NW Italy). The quartzite tower location is highlighted by the red rectangle. (**b**) Frontal view of the quartzite tower and active landslide at the base.

To characterize the stability of the tower and provide a continuous monitoring tool to the site, a network of four wireless passive monitoring stations was installed at the tower summit (S1 to S4 in Figure 2). Each station included a 2 Hz triaxial high-sensitivity geophone (200 V/(m/s)) and an on-purpose designed digitizer/recorder (GEA–GPS, developed by PASI s.r.l. and Iridium Italia s.a.s.) ensuring continuous seismic noise recording at 250 Hz sampling rate, low-weight of the instrumentation to reach remote areas, low power consumption in the absence of an external power supply (approximately 30 days of autonomy) and daily remote information about the system state of health by a GSM–GPRS module. Synchronization between the different stations is provided by GPS timing. Data are stored in 1 h files in an internal 32 GB memory card.

Stations S1 and S2 were installed in December 2019 at the top of the tower and on the stable rock mass. Stations S3 and S4 were added in March 2020 again on the tower top and at the back of the open fracture isolating the tower from the rock mass (Figure 2). This network configuration was chosen to obtain redundant information with two stations located at the top of the potentially unstable tower (S2 and S3) and two reference stations outside (S1 and S4). The monitoring network operated periodically until July 2020. The periods with lacking data are mainly due to COVID-19 travel restrictions in spring 2020, preventing battery replacement.

A geological survey was performed to analyze the fracturing of the site (Figure 2b). The quartzite tower is approximately 25 m high and 10 × 12 m wide at the top, with a visible set (K1) of near-vertical rear fractures oriented 310/75 (average dip direction/dip). The major fracture of this set (red in Figure 2c) is considerably open (50 cm wide approximately at the tower top) and persistent at depth, at least for the first 15–20 m from the top, above the overburden. This fracture is partially filled by fine-grained debris and loose earthy materials. The tower sides are bordered by an approximately orthogonal set of near-vertical fractures (K2, 235/80). Very steep gullies are generated by K2 set on the tower sides. The layering of the quartzite generates a third set of near-horizontal discontinuities (K3, 110/25), with decimeter-thick beds. The tower base lays on this set, gently dipping towards the back rock mass. The on-site measurements of fracture orientations are in agreement with the ones previously reconstructed by means of a remotely piloted aircraft system and digital photogrammetry for the whole cliff [16]. Given the steep site morphology and the poor

visibility of the tower due to the surrounding vegetation, the geological measurements carried out on-site coupled with GNSS (Global Navigation Satellite Systems) measurements of the accessible tower edges, were used as support (3D numerical modeling) for the tower geometry reconstruction.

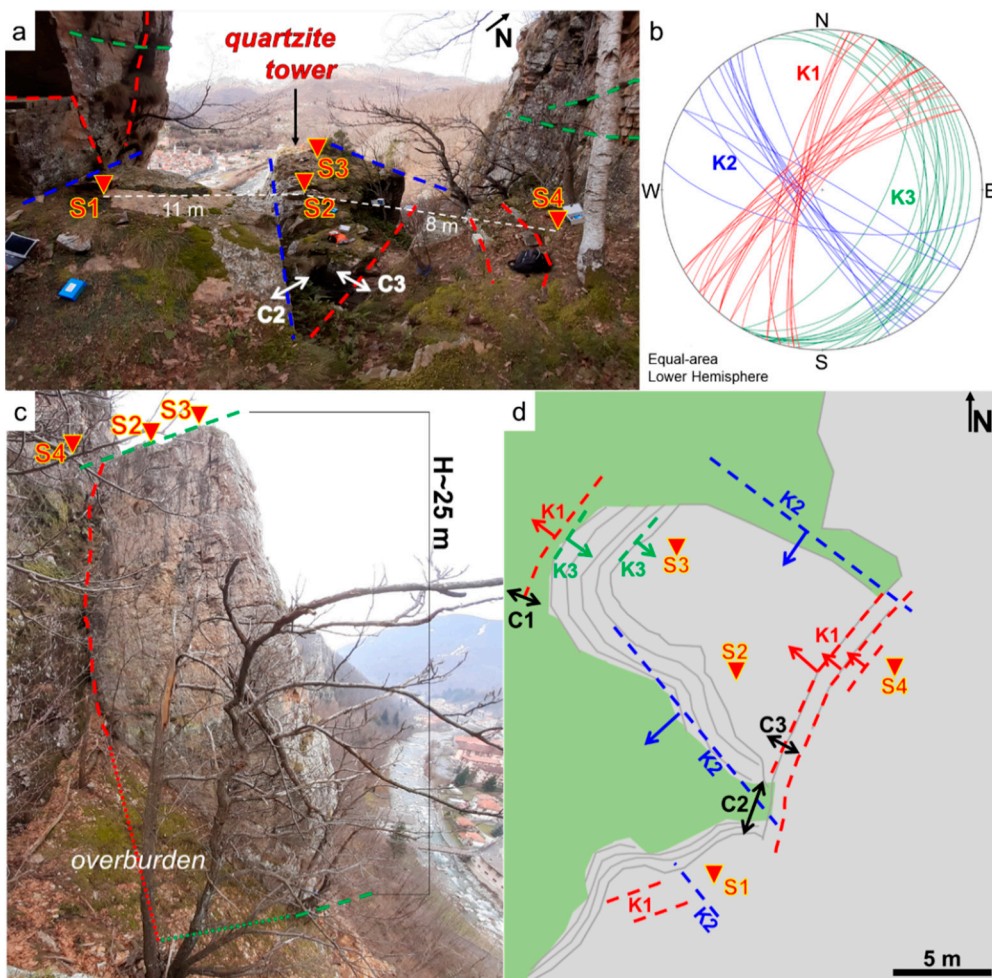

**Figure 2.** Geometry of the study site and monitoring network (**a**) Wide-angle view of the top of the quartzite tower (S1 to S4: passive seismic stations; C1 to C3: extensometers). (**b**) On-site measured orientations of fracture planes. (**c**) Side view of the tower. (**d**) Simplified planar view of fracture and sensor locations. In all panels, main fracture directions are highlighted by colored dashed lines (the arrows mark the dip direction). Three fracture sets were identified (dip direction/dip): K1 (310/75, in red); K2 (235/80, in blue); K3 (110/25; in green).

## 3. Materials and Methods

### 3.1. Ambient Seismic Noise Analyses

Ambient seismic noise spectral analyses and cross-correlation were computed on the acquired data set to characterize the seismic response and the stability of the tower and to monitor possible modifications in the seismic parameters with time driven by meteorological factors.

First, hourly Power Spectral Densities (PSDs) were computed for each component of the four monitoring stations [12,18,19] and compared with the New High and Low Noise Models (NHNM, NLNM) boundaries [20], for a global estimate of the recorded data quality. Probability Density Functions (PDFs) and time evolution of the computed PSDs were analyzed to recognize spectral amplifications in narrow frequency bands varying with time for the stations placed on the tower (S2 and S4), potentially representing natural resonance frequencies of the site.

Spectral ratio computations were performed to enhance the presence of these frequency peaks in the spectral content [5,7,12], both on single stations (e.g., H2/V2: horizontal-to-vertical spectral ratio on S2) and site-reference configurations (e.g., H2/H1: horizontal-to-horizontal spectral ratio between S2 and S1). For each station, the horizontal spectrum H was computed as total horizontal energy, by the composition of E and N hourly spectra.

Spatial directivity and amplitude of the spectral peaks were further investigated in comparison with the orientations of the fractures bounding the tower. The azimuth of vibration in the horizontal plane of the 2–10 Hz frequency components was calculated from the composition of the hourly E and N spectra of each station.

A 3D numerical model of the tower was built in Comsol Multiphysics to quantitatively validate the experimental resonance frequency values and vibration orientations and to deepen the investigation on the tower constraints through an eigenfrequency simulation [11]. A free tetrahedral mesh was applied to the simplified geometric model of the cliff. Fixed constraints were imposed only at the tower base, while all the other model boundaries were left free to vibrate. No in situ measurements of quartzite dynamic characteristics were available. Classical average parameters were adopted (i.e., density = 2550 kg/m$^3$, Poisson's ratio = 0.25, [21]). The P-wave velocity of quartzite was measured in the range of 3800 to 4800 m/s in experimental studies [22], which correlated with dynamic Young's modulus values ($E_{dyn}$) between 25 and 50 GPa. The lower boundary of this range corresponds to a number of discontinuities per meter equal to 10, that can be representative of the quartzite bed thicknesses measured on site.

Standard methods for ambient seismic noise cross-correlation were applied [5,12,23,24]. Cross-correlation was performed in the 2–20 Hz frequency range of vertical components. Hourly cross-correlograms were computed between site-reference stations (e.g., V2-V1) and filtered in 2 Hz bands. The hourly velocity change (dV/V) with respect to the average of all the considered correlograms was computed by the stretching technique [25,26] using time intervals [−2.5 s −0.5 s] and [0.5 s 2.5 s] to work on coda waves and avoid direct arrivals between the stations.

### 3.2. Microseismicity Analyses

Short-duration, high-energy events were extracted from the ambient seismic noise dataset with a common STA/LTA (Short Time Average over Long Time Average) algorithm. Variable combinations of STA/LTA parameters were tested on an initial time window of 1-week length to ensure the optimal set up for the detection of a wide data set of significant seismic events (STA window = 0.3 s; LTA window = 30 s; STA/LTA threshold = 6). The classification of the detected events was then performed integrating visual analysis of the events spectrograms [27] and cluster analysis on six salient time- and frequency-domain event parameters (I–VI in the following) [28]. Bracketed duration (i.e., the time interval between the first and last exceedance of a threshold equal to 15% of the signal maximum amplitude, I) and uniform duration (i.e., the sum of the single time intervals above the same threshold, II) were computed to account for multiple repetitions of short events in the same detected time window. The maximum amplitude of the signal normalized to the average amplitude computed over the event duration ($A_{max}/A_{mean}$, III) was considered to distinguish short-impulsive events from longer events without clear peaks in amplitude. Event kurtosis (IV) was computed to have an additional measure of the signal sharpness. The kurtosis is a statistical value characterizing the shape of a given amplitude distribution (e.g., normal distributions have kurtosis of 3). In seismological and seismic applications, extremely high kurtosis values can be related to sharp impulsive seismic signals, while low kurtosis values are expected in the background ambient seismic noise and in non-impulsive signals [29,30]. The peak frequency (V) was automatically picked from the event amplitude spectrum. The latter was also segmented in 5 Hz frequency classes (5 to 100 Hz), to further identify the frequency range having maximum amplitude (VI). These 5 Hz classes were considered to obtain redundant spectral information and reduce the effect of possible spectral spikes on the peak frequency determination.

First, the visual classification of an initial data set of 158 event spectrograms recorded at S2 was carried out. This step led to the recognition of 50 microseismic events (MS), 50 long-duration microseismic events (LONG-MS), 50 low-frequency events (LOW-F) and 3 regional earthquakes (EQ). Rare signals containing electromagnetic noise only, probably linked to the station itself, were also identified and included in the initial set (5 events, EM-NOISE).

Secondly, k-means cluster analysis [31] was undertaken on the six classification parameters computed on the visually classified events. This clustering scheme was chosen since the only input needed is the desired number of clusters. For each cluster, the six-dimension (6 classification parameters, I–VI) centroid is computed. Each signal is then automatically assigned to the cluster having the nearest centroid. Starting with an initial number of 5 clusters (i.e., the number of visually identified event classes), with a trial-and-error procedure we increased the number of clusters up to 16 subclusters (2 for EM-NOISE, EQ, 4 for MS, LONG-MS and LOW-F) to completely match the visual classification in 5 classes. The 6 classification parameters were then computed for the whole dataset and on all stations. The 6-dimension distance between each event parameters and the 16 centroids was evaluated and each event was automatically assigned to the subcluster with the nearest centroid, to eventually obtain the 5 main clusters of events.

A random visual check on approximately 100 automatically classified events gave fully satisfactory results. The temporal trend of the natural events possibly related to fracturing or slip movements and the related released seismic energy [32] were then computed and analyzed in comparison with the meteorological parameters of the site and the available displacements measurements.

## 4. Results

### 4.1. Ambient Seismic Noise Spectral Analyses

Ambient seismic noise results are discussed in the following for the stations S1 (outside the tower) and S2 (tower top) having the longest recordings (December 2019–June 2020) and high quality data on all components. However, S3 and S4 stations (not shown) exhibited spectral responses very close to S1 and S2 respectively.

Examples of representative PDFs and time evolutions of PSDs are shown in Figure 3 (S1) and Figure 4 (S2) in the 2–20 Hz frequency range (above the central frequency of the adopted geophones). The spectral response of all channels falls within the NLNM and NHNM boundaries [20]. Ambient seismic noise recorded at S2 shows greater amplification with respect to S1 in two frequency bands located around 6 Hz (f1 in Figures 3 and 4) and between 8 and 10 Hz (f2 in Figures 3 and 4). All components of S2 show marked amplifications around f1. A weaker peak in the same frequency range is noticed only on N1 (Figure 3b). Therefore, S2 can be considered as a reference stable station even if some influences due to its proximity to the tower are present in the data. Both f1 and f2 vary with time and generally increase over the monitored period. Due to the above considerations, f1 and f2 can be considered as the first two resonance frequencies of the quartzite tower. A third spectral amplification may be weakly present at both stations in the 10–15 Hz frequency band (f3 in Figures 3 and 4).

Both single-station (e.g., H2/V2, Figure 5a,c) and site-reference (e.g., H2/H1, Figure 5b,d) spectral ratios confirm and emphasize these spectral observations, with a better separation between f1 and f2 in site-reference spectral ratios. H2/V2 spectral ratio is close to 10 at f1 and f2, but amplitude variations with time are depicted on both frequencies. The general frequency increase with time is confirmed by both spectral ratios (Figure 5c,d).

The spectral amplitude and azimuthal orientation in the horizontal plane of f1 and f2 was consequently further investigated to analyze the vibration modes of the quartzite tower in comparison with the fracture systems bordering the potentially unstable volume.

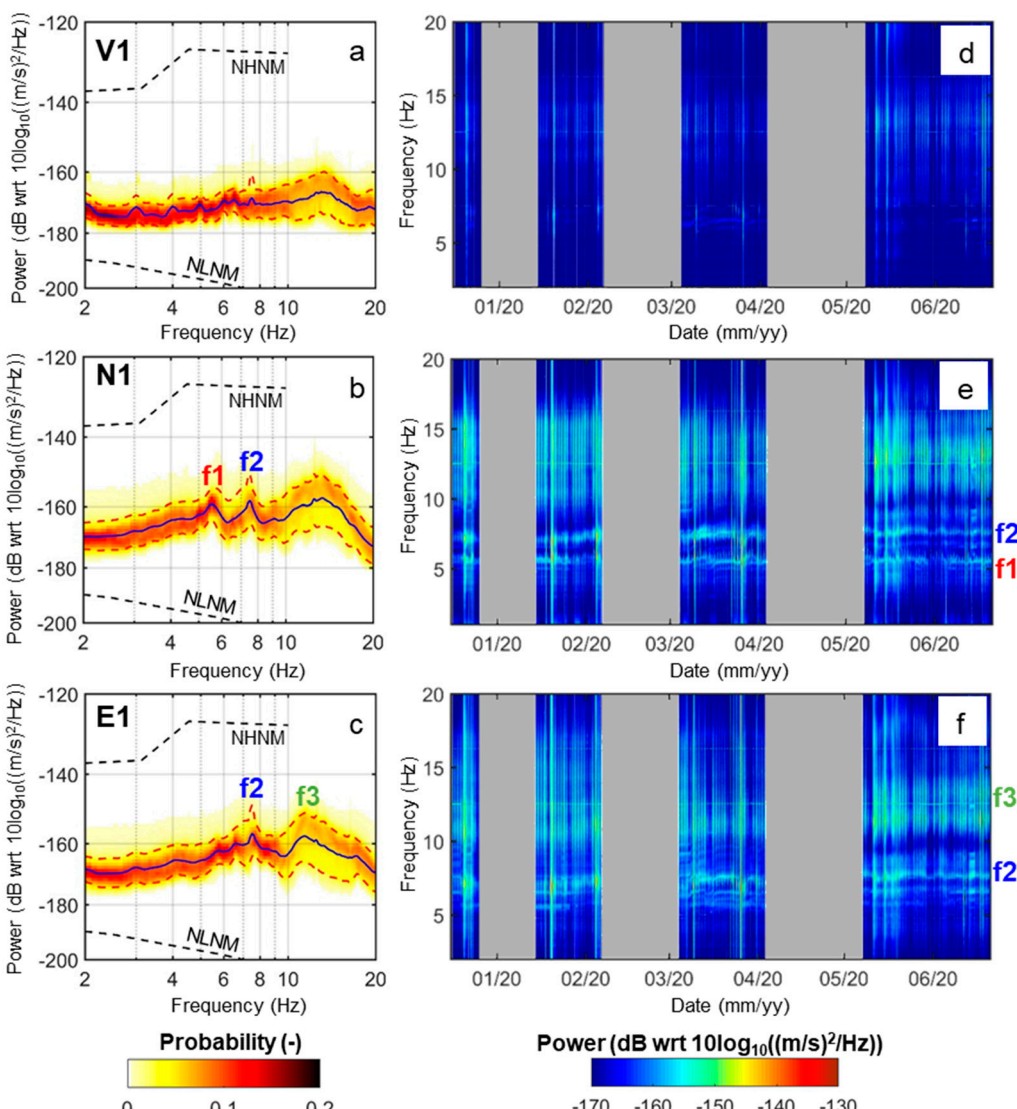

**Figure 3.** Spectral analysis of ambient seismic noise recorded at S1 (reference station). (**a–c**) PDFs of the hourly PSDs, with average (blue line) and 10th–90th percentiles (red dashed lines): (**a**) Vertical, (**b**) North and (**c**) East component. Black dashed lines: NHNM and NLNM boundaries [20]. (**d–f**) Hourly PSDs over the monitored period: (**d**) Vertical, (**e**) North and (**f**) East component. The main spectral peaks are highlighted with f1 to f3. No data periods are shaded in gray.

### 4.2. Experimental and Modeled Ground Motion Orientation

An exemplificative experimental polar plot of f1 and f2 vibration directions recorded during one hour, at top of the potentially unstable tower (S2) is shown in Figure 6a. The orientation of the two frequencies is almost perpendicular, with f1 vibrating in NW-SE direction and f2 oriented NE-SW. As shown for f1 and f2 values (e.g., Figure 5c,d), also azimuths of vibration and spectral amplitudes are not constant in time. The vibration orientation is almost stable for f1 (45° from N direction, Figure 6c), while f2 shows azimuthal fluctuations over time (Figure 6d) in the 125–150° range from N direction. A clear decrease in the azimuth is noticed in mid-March 2020. This decrease in concomitant to an increase in f2 value (Figure 6a). The spectral amplitude of f1 (Figure 6e) is greater than the one of f2 during the winter months at the beginning of the monitored period (e.g., Figure 6a), while the two spectral amplitudes become roughly comparable from May 2020.

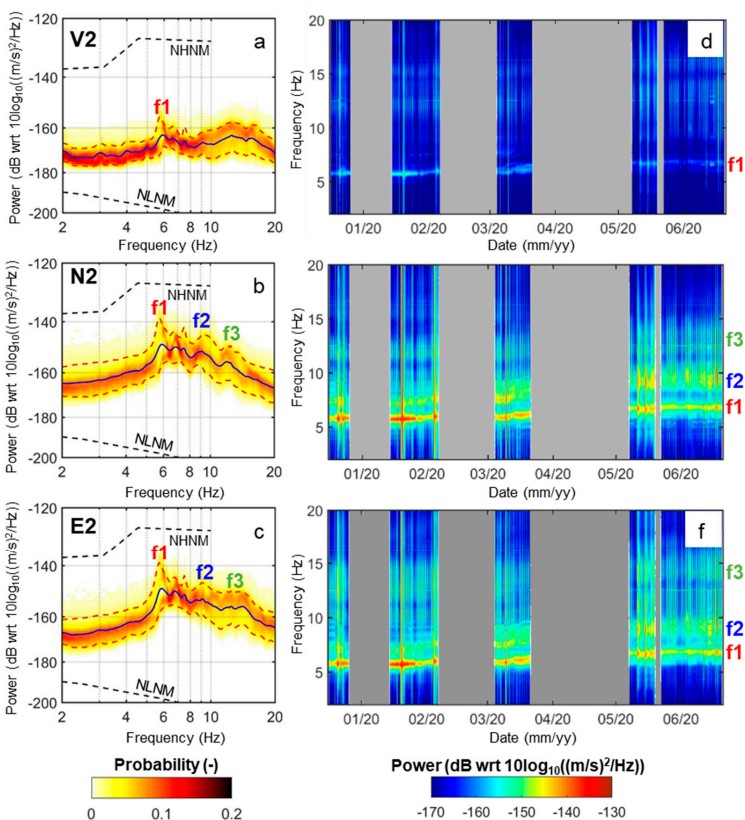

**Figure 4.** Spectral analysis of ambient seismic noise recorded at S2 (tower top). (**a–c**) PDFs of the hourly PSDs, with average (blue line) and 10th–90th percentiles (red dashed lines): (**a**) Vertical, (**b**) North and (**c**) East component. Dashed black lines: NHNM and NLNM boundaries [20]. (**d–f**) Hourly PSDs over the monitored period: (**d**) Vertical, (**e**) North and (**f**) East component. The main spectral peaks are highlighted with f1 to f3. No data periods are shaded in gray.

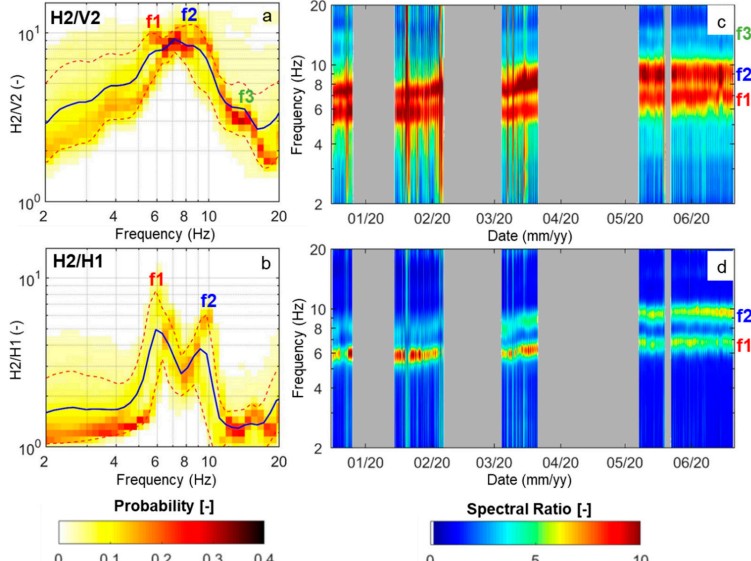

**Figure 5.** Ambient seismic noise spectral ratios. PDFs of the hourly computed (**a**) H2/V2 and (**b**) H2/H1 spectral ratios, with average (blue line) and 10th–90th percentiles (dashed red lines). Temporal evolution of (**c**) H2/V2 and (**d**) H2/H1 in the monitored period. The main spectral peaks are highlighted with f1 to f3 (on the right). No data periods are shaded in gray.

The first vibration mode (f1) is nearly perpendicular to the fractures of K2 set, laterally delimiting the tower, while the second vibration mode (f2) is perpendicular to the rear fracture of K1 set (compare Figures 1d and 6a). These vibration orientations are unusual with respect to literature studies on nearly 2D rock columns [5–10], for which the first vibration mode is always found to be perpendicular to the back fractures, thus indicating the potential direction of collapse.

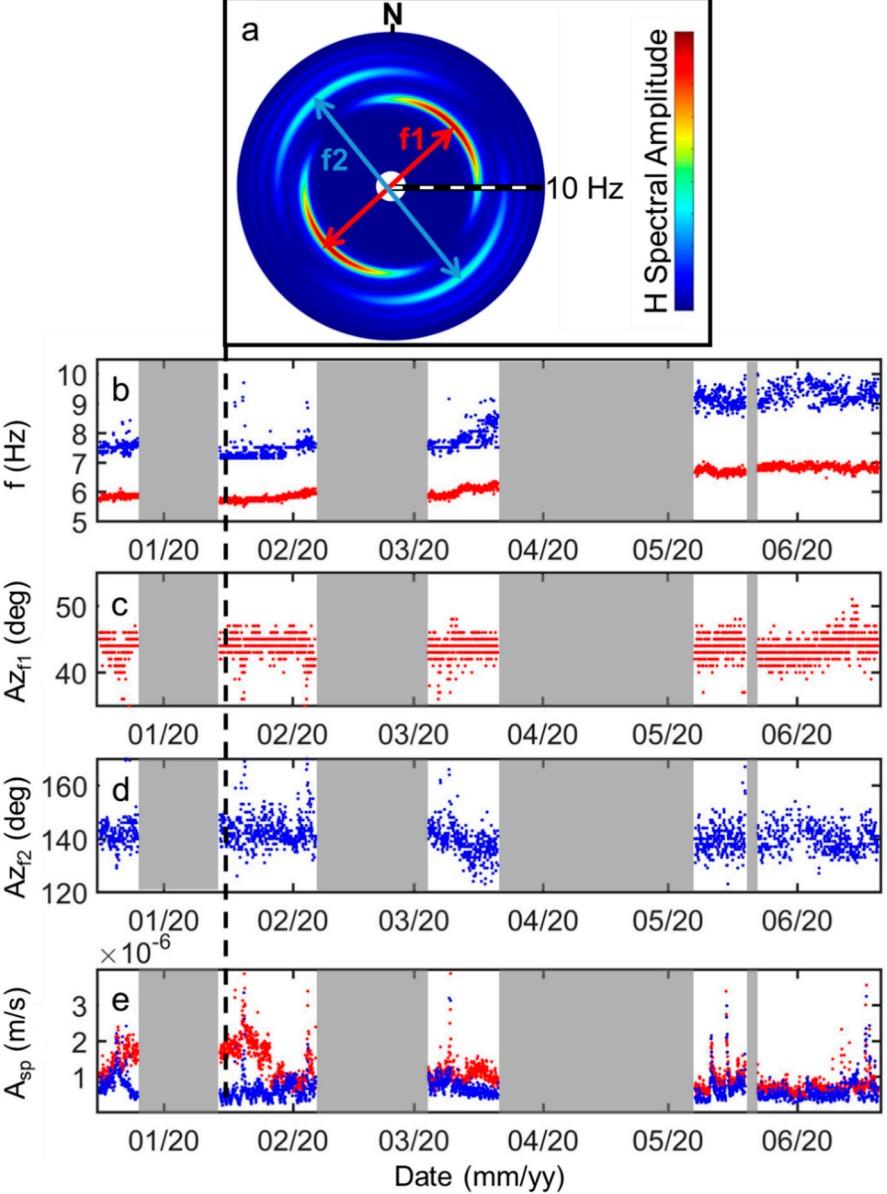

**Figure 6.** (**a**) Exemplificative polar plot of S2 horizontal spectral components computed on a 1 h noise recording (the location in time in highlighted by the dashed black line in the following panels). (**b–e**) Temporal evolution of (**b**) values, (**c–d**) azimuths (in degrees from N), (**e**) spectral amplitudes of the first two resonance frequencies (f1 in red, f2 in blue) measured on the horizontal plane at S2. No data periods are shaded in gray.

The 3D numerical modeling of the tower's seismic response was therefore undertaken to validate the experimental outcomes (Figure 7). Several tests were carried out, varying the geometrical constraints at the base of the tower and along the rear fracture of K1 set. If a rock bridge is imposed along the rear fracture close to the base of the tower, even for a few meters only in height, the first modeled vibration mode is bending perpendicular to

K1 fracture direction, as usually found in literature studies. The only way to reproduce the experimental vibration orientations is to constrain only the tower base.

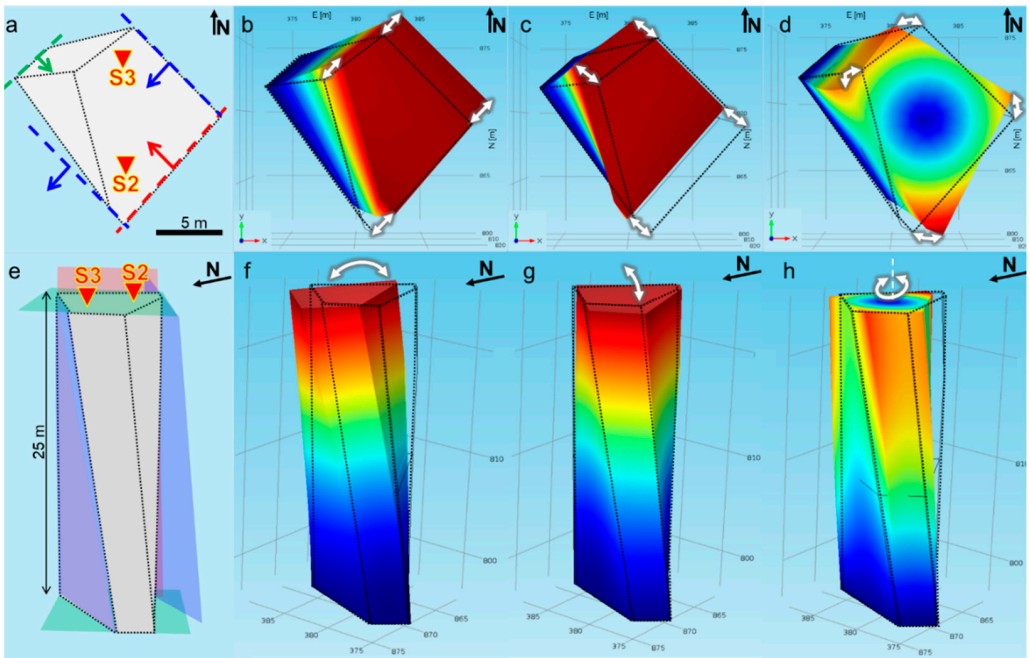

**Figure 7.** 3D numerical model of the quartzite tower. (**a–d**) planar view, (**e–h**): frontal view. (**a,e**) Initial model geometry with indication of the delimiting fracture orientations (K1 in red, K2 in blue, K3 in green) and sensor locations (S2 and S3). (**b,f**): First, (**c,g**): second, (**d,h**): third vibration mode. White arrows mark the vibration direction from the undeformed volume (dashed black contour). The color scale (from blue to red) indicate increasing amount of displacement.

In this configuration, the experimental resonance frequency vibration orientations are numerically reproduced with two bending modes perpendicular to K2 (f1) and K1 (f2) directions.

Experimental and modeled results are compared in Table 1 for the value and azimuth of f1 and f2, considering the lowest experimental estimation of the quartzite Young's modulus ($E_{dyn}$ = 25 GPa), suitable for a rock mass with 10 discontinuities/m [22], corresponding to the average bed thickness generated by K3 set.

**Table 1.** Experimental and modeled resonance frequencies f1 and f2 and related average azimuths of vibration from N direction (quartzite Young's modulus = 25 GPa).

|  | f1 (Hz) | f2 (Hz) | $Az_{f1}$ (°) | (°) |
|---|---|---|---|---|
| **Measured** | 5.2–6.9 | 7.2–9.4 | 45 | 140 |
| **Calculated** | 5.4 | 7.0 | 40 | 135 |

In the numerical simulation, a third vibration mode is obtained at higher frequency (f3 = 17.3 Hz), corresponding to torsion of the tower around its vertical axis. This vibration mode shows very low amplitude of vibration at the tower top in the numerical results (Figure 7d,h), and may reflect the weak spectral peak found in the 10–15 Hz frequency range in the experimental data (e.g., Figure 5c).

A sensitivity analysis on the quartzite Young's modulus estimation was also carried out. Increasing $E_{dyn}$ up to 40 GPa did not modify vibration orientations but led to an increase in resonance frequency values (f1 = 6.8 Hz, f2 = 8.9 Hz, f3 = 21.8 Hz), still in the range of the experimental measurements during warm months. $E_{dyn}$ of 50 GPa (experimental upper limit for massive quartzite, [22]) led to frequency values outside the

experimental range (f1 = 7.6 Hz, f2 = 9.9 Hz, f3 = 24.33 Hz), thus confirming the role of fracturing conditions on the quartzite mechanical properties.

### 4.3. Ambient Seismic Noise Cross-Correlation

Cross-correlation results are shown in Figure 8 for the station pair S2–S1 in two successive 2 Hz frequency bands, almost overlapping f1 (5–7 Hz) and f2 (7–9 Hz). Correlation coefficients (CCs) are higher for the velocity change (dV/V) estimated in the 5–7 Hz frequency band (Figure 8d). A negative velocity change (−5%) is depicted from December 2019 to mid-January 2020 in this frequency band (Figure 8c), associated with a temporary drop in CC values, and followed by a progressive increase in dV/V in early spring months. The velocity remained almost constant over the last two monitored months.

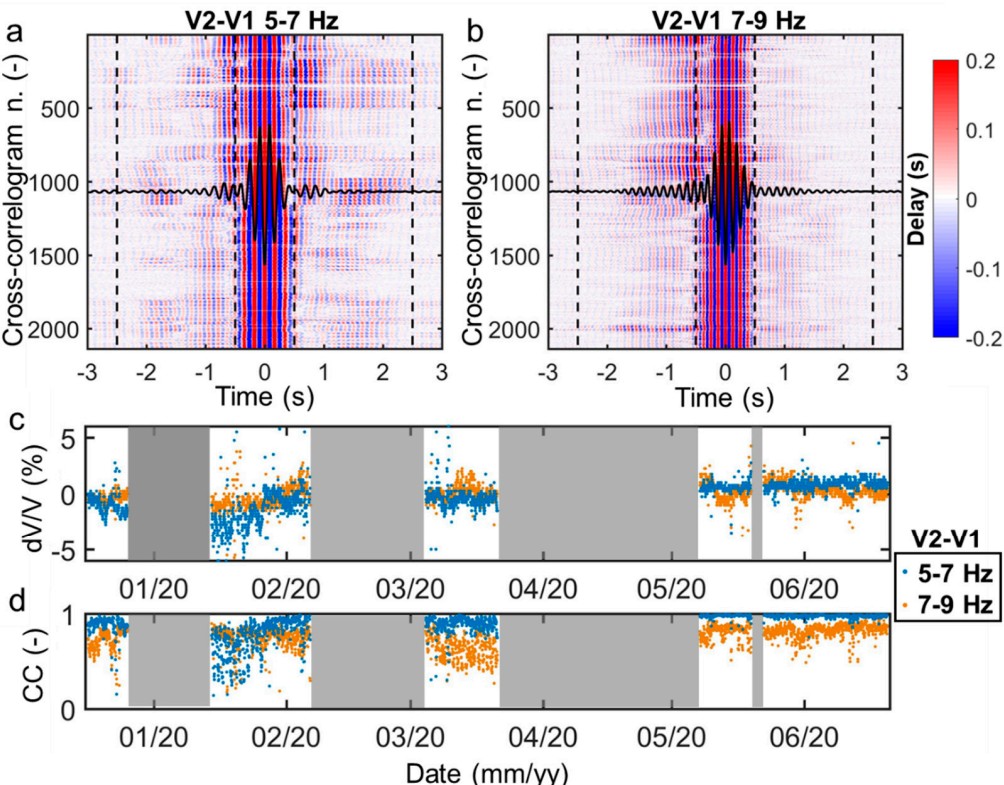

**Figure 8.** Cross-correlation of ambient seismic noise recorded at S2 and S1 (vertical components). (**a**,**b**): hourly cross-correlograms in the (**a**) 5–7 Hz and (**b**) 7–9 Hz frequency bands. The reference correlogram (average of the correlograms shown in each panel) is displayed in the center by the continuous black line. Dashed vertical black lines delimit the time intervals [−2.5 s −0.5 s] and [0.5 s 2.5 s] used for velocity change (dV/V) estimation. (**c**) Obtained hourly dV/V and (**d**) related correlation coefficients in the two frequency bands. No data periods are shaded in gray.

### 4.4. Microseismicity Analyses

The analysis of the seismic signals extracted from the continuous ambient noise recordings led to the recognition of four classes of natural events (Figure 9).

Short-duration events with an almost triangular envelope and a sharp high-frequency emerging onset, followed by a sudden exponential decay of the high-frequency content, were related to micro-cracking and micro-fracturing processes within the investigated rock mass (Figure 9a–c). These microseismic events (MS) showed spectral features fully comparable with microfracturing signatures reported in the literature studies on different rock types at both laboratory and field scales [15,28,33,34].

A second class of events (Figure 9d–f) showed spectral features similar to the microseismic events but longer duration in time domain. Consequently, we interpreted

these signals as trains or long sequences of repeated microseismic events generated by high-energy long-lasting microfracturing or sudden slip processes (LONG-MS).

A third class of events (Figure 9g–i) exhibited long duration, comparable to the LONG-MS events, but a significantly lower frequency content (LOW-F) and slowly emerging signals. These signals could be possibly related to slip along pre-existing fractures and be promoted by water infiltration.

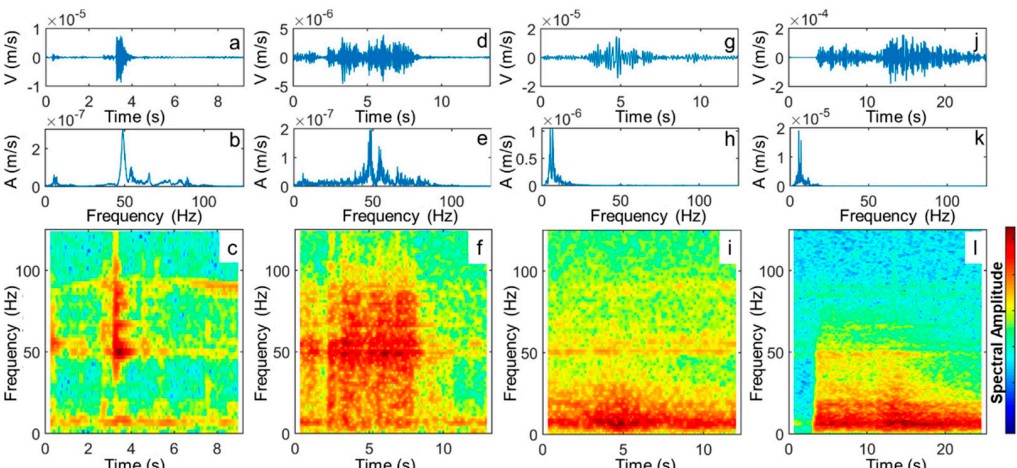

**Figure 9.** Example of seismic events representative of the four classes of signals generated by natural sources. (**a–c**): Microseismic events due to micro-fracturing processes (MS). (**d–f**): Long-lasting sequence of microseismic events (LONG-MS). (**g–i**): Low-frequency events possibly related to slip processes on pre-existing fracture surfaces. (**j–l**): earthquake (19 January 2020, 05:22:06.2, 44.72°N 8.1°E, depth = 12 km, $M_L$ = 3.1). In each column, from top to bottom: seismogram recorded on channel V2 (S2), amplitude spectrum and spectrogram.

These three classes were thus related to possible fracturing and slip phenomena potentially affecting the tower stability. A few local and regional earthquakes were also recorded by the network in the monitored period, showing very long duration and low frequency content (<10 Hz) and peculiar spectral signature (EQ, Figure 9j–l). A fifth class of events not related to natural sources but likely cause by electromagnetic disturbances on the network instrumentation was finally identified (EM-NOISE, not shown), but easily removed from the global data set because of the different time- and frequency-domain features.

Examples of the classification parameters adopted for the automatic clustering and classification are shown in Figure 10a,b. MS events showed lower bracketed and uniform duration with respect to LONG-MS and LOW-F events, higher amplitude and kurtosis values due to their sharp onset. LONG-MS and LOW-F signals, almost overlapped in Figure 10a,b were mainly differentiated for their different spectral content, as shown in Figure 10c. Rare EQ events were automatically detected mainly for their low frequency content, while they showed kurtosis, amplitude and duration in the same range of many LONG-MS and LOW-F events. EM-NOISE signals clustered at very low kurtosis and amplitude values.

In Figure 10c–e the time evolution of the events possibly related to the tower stability is further investigated. The time evolution of the peak frequency of MS, LONG-MS, LOW-F event recorded at S2 is shown in Figure 10c. LOW-F events cluster at frequencies lower than 15 Hz and their rate increase in the last two months of recordings. LONG-MS and MS peak frequencies overlap in two separated frequency ranges centered around 50 Hz and 55 Hz. Rarely, events with higher peak frequencies are detected. The daily event rate of these three classes and cumulative event number is reported in Figure 10d for S2, in comparison with the event rate recorded at S1 (Figure 10e).

The total number of events recorded by each station of the monitoring network is summarized in Table 2.

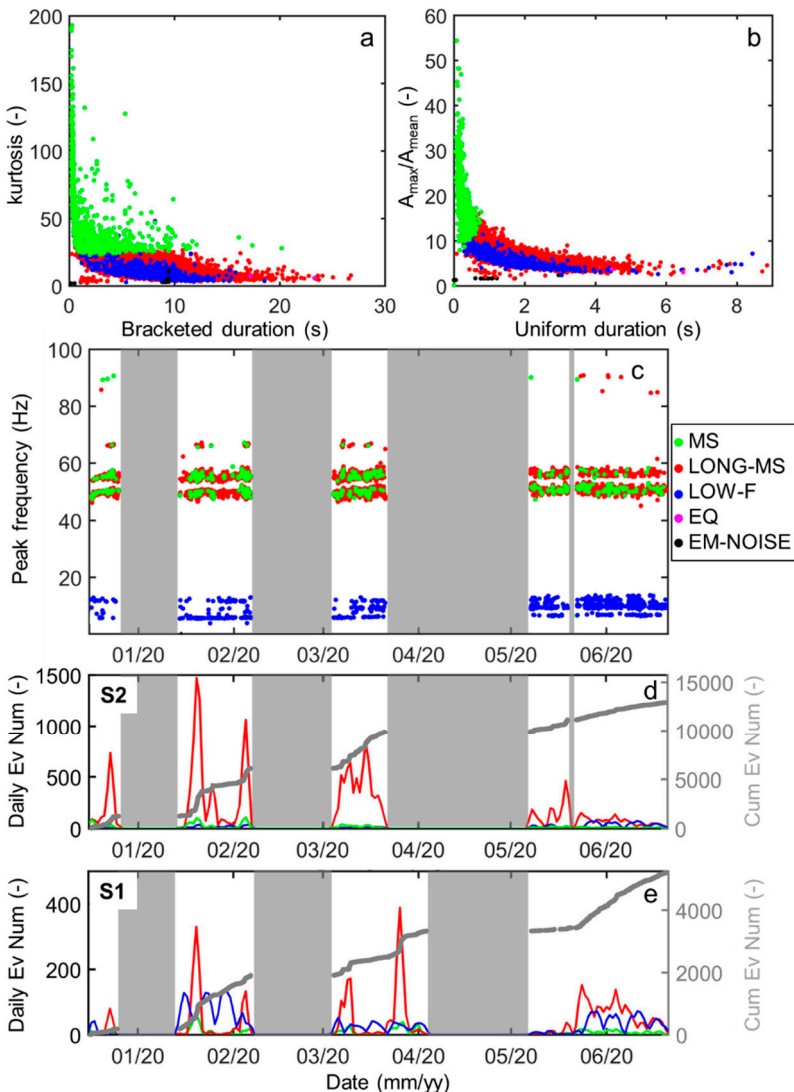

**Figure 10.** Automatically classified seismic events derived from the ambient seismic noise data set (MS in green, LONG-MS in red, LOW-F in blue, EQ in magenta, EM-NOISE in black). (**a**) kurtosis vs. bracketed duration plot of the classified events. (**b**) Maximum amplitude of the signal normalized to the average value computed over the whole signal duration plotted in comparison with the event uniform duration. (**c**) Peak frequency evolution in time of MS, LONG-MS, LOW-F events. (**d–e**) Daily and cumulative event number of MS, LONG-MS, LOW-F events recorded at (d) S2 (tower top) and (**e**) S1 (reference station). The y-scales in (**d**) are three times bigger than the ones in (**e**). No data periods are shaded in gray for each station.

**Table 2.** Number of events recorded at the four monitoring stations.

|  | **S1** | **S2** | **S3** [1] | **S4** [1] |
|---|---|---|---|---|
| MS | 480 | 898 | 675 | 671 |
| LONG-MS | 2623 | 10,891 | 3240 | 1619 |
| LOW-F | 2086 | 1090 | 620 | 635 |
| EQ | 5 | 5 | 2 | 2 |
| NOISE | 12 | 24 | 1 | 3 |
| *Total* | 5206 | 12,908 | 4538 | 2930 |

[1] Shorter monitored period (March–July 2020).

　　The total number of events is considerably higher at the stations located on the quartzite tower (S2 and S3). Reference stations (S1 and S4) however recorded a higher

number of LOW-F events, while LONG-MS and MS are predominantly recorded at the top of the tower, supporting the hypothesis of a genesis related to the tower behavior.

## 5. Discussion

Both ambient seismic noise and microseismicity analyses carried out on the quartzite tower emphasized seismic features related to the stability of the investigated volume.

Ambient seismic noise spectral analysis highlighted amplification in two distinct frequency bands (f1 and f2) at the stations located at the tower top, almost absent at the reference stations located outside the volume, and interpreted as the first two resonance frequencies of the tower. The vibration orientations at f1 and f2 were found to be controlled by the two main fracture sets (K1 and K2) delimiting the potentially unstable volume. In contrast to previous case studies on rock columns and prisms [5–10], f1 vibration was not found to be perpendicular to the open rear fracture of K1 set, but indicated bending perpendicular to the lateral fractures of K2 set. The main causes of this unusual bending orientation were attributed to the tower size, with a longer extension in the direction of K2 (approximately 12 m) with respect to the width in K1 direction (approximately 10 m) and the basal dip and dip direction towards the back slope (K3 set). The experimental findings were confirmed by 3D numerical modeling. The latter gave additional information on the tower rear constraints, partially hidden by the overburden on the tower sides. To reproduce the experimental vibration orientations, the tower needed to be modeled without any constraint at the rear fracture, thus probably indicating fracture opening down to the tower base and the absence of rock bridges between the tower and the stable cliff.

To fully understand the detected seismic response, ambient seismic noise spectral analyses, cross-correlation results and microseismicity trends were compared with air temperature, and the precipitation is measured at the nearest ARPA Piemonte meteorological station (5 km SW of the site) and with displacement measurements (Figure 11).

Resonance frequencies variations at the seasonal scale were visibly controlled by temperature: f1 and f2 values increase as air temperature increases at the end of winter months (Figure 11b). Velocity changes (dV/V) detected by ambient seismic noise correlation generally followed the same seasonal temperature-driven trend (Figure 11c). A positive correlation between T and the two monitored parameters was established even if the monitored period was too short to estimate the seasonal delay in temperature modifications. This temperature-driven mechanism, which induces resonance frequency and velocity change-reversible variations had been previously identified in several case studies (e.g., [5,7,12,15]) and classified as "fracture effect" [1]. Air temperature fluctuations induced thermal expansion and contraction on the quartzite tower. With increasing temperature at the end of the winter period, tower thermal dilation caused progressive closing of the fractures and microcrack, as testified by the decreasing displacements across the fractures at the tower top (C2 and C3 in Figure 11g) and by the variations in the vibration azimuth depicted on f2 (Figure 6d). A relative increase in fracture contact stiffness generated an increase in resonance frequency values and seismic velocity. The same temperature-driven fluctuations were depicted in the peak frequencies of the recorded microseismic events (MS and LONG-MS, Figure 11d). Interestingly, the peak frequencies cluster in two separated frequency ranges, probably suggesting the existence of two different and recurrent sources of these events, both sensitive to the air temperature variations affecting the mechanical response of rock mass and fractures. However, the event source location is limited by the network geometry and by the absence of a detailed 3D velocity model for the tower. These limitations did not allow for confirmation of this hypothesis and further locating of the two clusters.

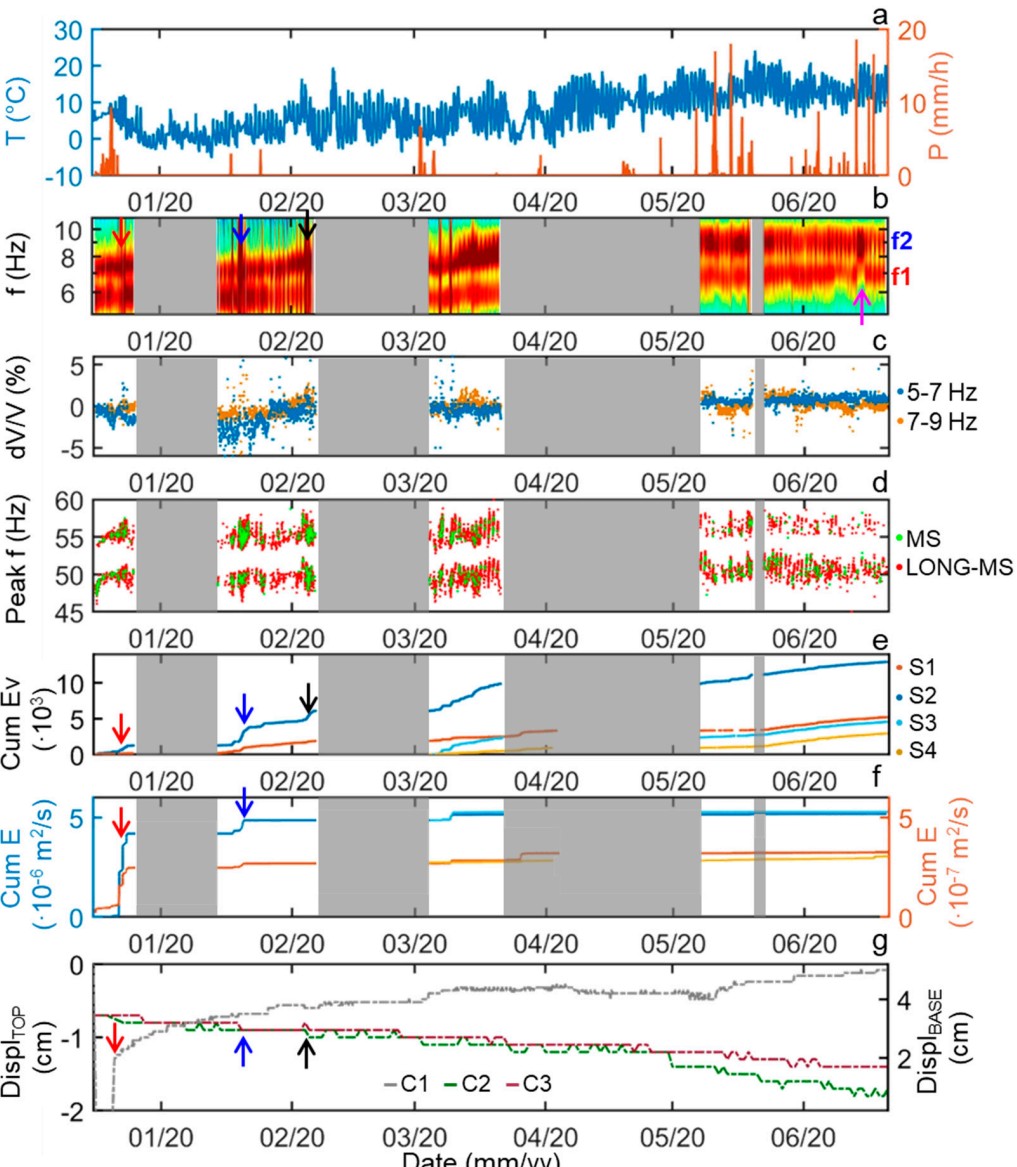

**Figure 11.** Comparison of ambient noise and microseismicity results with meteorological variations and displacement measurements. (**a**) air temperature variations (blue) and hourly precipitation amount (orange) at the nearest ARPA Piemonte meteorological station (Ormea–Ponte di Nava). (**b**) Zoom on H2/V2 spectral ratio around the first two resonance frequencies (same color scale of Figure 5c). (**c**) Seismic velocity changes retrieved from ambient noise cross-correlation in the frequency bands overlapping f1 and f2. (**d**) Peak frequency of the microseismic events recorded at S2. (**e**) Cumulative number of tower-related (MS, LONG-MS, LOW-F) events recorded at the four monitoring stations. (**f**) Related cumulated seismic energy. To improve the comparison, the cumulated energy of S3 and S4 (operating from March 2020) is plotted starting from the cumulated energy value of S2 and S1 respectively at their first day of recording. (**g**) Displacement measurements at C1 (tower base), C2 and C3 (tower top). No data periods for S2 are shaded in gray. Colored arrows mark salient windows in which the seismic response is not correlated to air temperature modifications: spectral content perturbations and high microseismicity after precipitations or concomitant to displacement (e.g., in red, blue, black), f1 spectral amplitude decrease after intense precipitation (in magenta).

At a daily scale (Figure 12), positive correlation between temperature and f2 values was also found, with a delay of approximately 12 h between air temperature and frequency variations (Figure 12b). Despite some noise in the results, the same response and delay were highlighted by dV/V fluctuations computed in the frequency band of f2 (7–9 Hz,

Figure 12c). In contrast, f1 seemed poorly sensitive to daily temperature fluctuations, with a vibration azimuth remaining almost constant in the monitored period (Figure 6c). Seismic velocity changes measured in the 5–7 Hz frequency band and peak frequencies of the microseismic events did not show any delay to air temperature variations. These results highlighted a different behavior along the K1 and K2 fractures resulting from a different degree of freedom along the two sets. The tower sides (K2 set) were indeed directly exposed to air temperature fluctuations and free to vibrate, while the rear fracture (K1 set) was more constrained, partially filled by debris and not directly exposed to solar radiation.

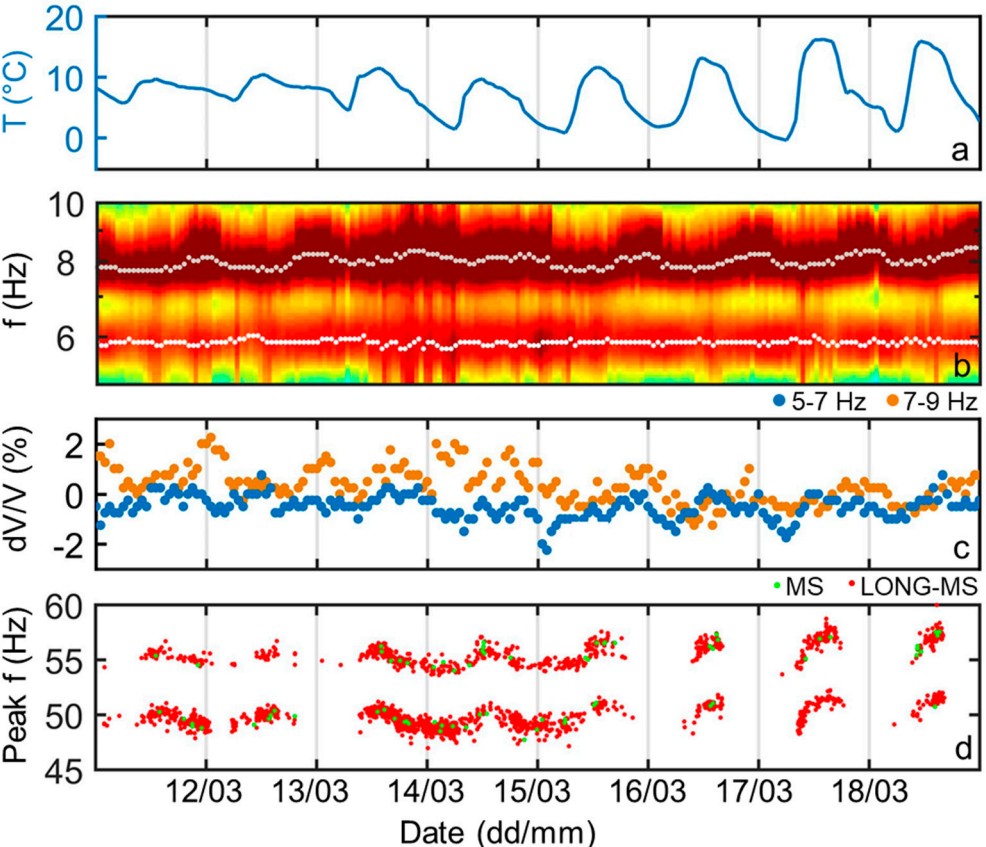

**Figure 12.** Zoom on daily seismic parameters variations in comparison with air temperature over an 8-day window (Mid-March 2020) without precipitation. (**a**) air temperature variations at the nearest ARPA Piemonte meteorological station of Ponte di Nava. (**b**) Zoom on H2/V2 spectral ratio around the first two resonance frequencies (same color scale of Figure 5c) with hourly f1 and f2 maxima highlighted by white and gray dots. (**c**) Seismic velocity changes retrieved from ambient noise cross-correlation in the frequency bands overlapping f1 and f2. (**d**) Peak frequency of the microseismic events recorded at S2.

No obvious influence of precipitation was noticed on the resonance frequency values. However, a clear decrease in f1 spectral amplitude was depicted immediately after the highest precipitation peak (18 mm/h in June 2020, Figure 11b, magenta arrow). The heavy rainfalls of December 2019 (red arrow) triggered a significant opening of the fracture at the base of the tower, as well as a dramatic increase in microseismicity. A clear perturbation in the ambient noise spectral content was also found in correspondence. Later increases in the number of microseismic events were associated with less marked increases in released seismic energy and likely associated with displacements at the top of the tower. Perturbations in the ambient noise spectral content were depicted in the same time windows (e.g., blue and black arrows in Figure 11). More in general, steep increases in microseismicity were found until April 2020, in parallel with resonance frequency and velocity change

increases. In the following period, the microseismic activity became more constant, and high-energy MS and LONG-MS were mostly replaced by an increase in the number of low-energy LOW-F events (Figure 10d,e), possibly driven by the diverging displacements of the tower top and base fostered by increasing temperature and precipitation rates in May–June 2020. Water infiltration may indeed promote slip movements along pre-existing surfaces and result in these low-frequency seismic signatures [28].

## 6. Conclusions

The six-month monitoring experience on a quartzite tower highlighted the suitability of passive seismic monitoring systems for the characterization of potentially unstable sites and early warning purposes. No irreversible modifications uncorrelated with meteorological parameters indicating an acceleration towards the tower collapse, were recorded. Ambient seismic noise parameters showed reversible fluctuations mainly driven by air temperature. Precipitation amounts were not found to directly influence resonance frequency values and velocity changes, but temporary modifications in spectral amplitude were detected in periods with frequent or intense precipitation. A general change in the seismic event type, rate and released energy was identified in the same time windows. A clear match between the microseismic observations and the on-site measured displacement was found.

Despite the simple geometry of the tower, ground motion analyses and 3D numerical modeling highlighted a more complex vibration pattern with respect to the literature studies on nearly 2D rock columns and prisms, more similar to 3D case studies.

The joint interpretation of ambient noise and microseismicity data enabled to follow daily and seasonal modifications in the site response to external modifications. Further analyses are needed to fully understand the temperature-driven fluctuations of the peak frequencies of the detected microseismic events. Future perspectives of the work also include the reconstruction of a reliable 3D velocity model of the tower for microseismic event location. Active seismic surveys at the tower top are however limited by the reduced available space and the challenging site morphology and accessibility.

In similar future studies, ambient noise spectral amplitude and azimuthal variations should be further analyzed together with the resonance values for a global understanding of the site seismic response.

**Author Contributions:** Conceptualization, C.C.; methodology, C.C. and D.J.; software, C.C.; validation, C.C., D.J. and A.G.; investigation, C.C.; data curation, C.C.; writing—original draft preparation, C.C.; writing—review and editing, A.G., D.J. and C.C.; visualization, C.C.; supervision, A.G.; funding acquisition, A.G. and C.C. All authors have read and agreed to the published version of the manuscript.

**Funding:** This research received no external funding.

**Data Availability Statement:** The data presented in this study are available on request from the corresponding author.

**Acknowledgments:** We warmly thank our colleague Diego Franco, from the Applied Geophysics Lab of Politecnico di Torino, for his fundamental support in network installation and maintenance. We are also grateful to PASI s.r.l. and Iridium Italia s.a.s. for their great work and commitment in the development of the seismic instrumentation used for this study. In this respect, we particularly thank Lorenzo Bidone, Osvaldo Pirchio and Gabriele Calabrese for their dedication, patience and constant support in all the design, development, testing, deployment and maintenance phases. We are sincerely grateful to ARPA Piemonte for the permission to publish the monitoring displacement measurements and meteorological data, the fundamental help network deployment and site accessibility, the research discussion with Daniele Bormioli and Giuseppina Moletta.

**Conflicts of Interest:** The authors declare no conflict of interest.

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
