# Peer review of "Ambient Seismic Noise and Microseismicity Monitoring of a Prone-To-Fall Quartzite Tower (Ormea, NW Italy)"

_remotesensing, doi:10.3390/rs13091664_

Round 1

Reviewer 1 Report

This is an interesting paper studying the vibrational modes of a quartzite tower.  It is well written.  I am not familiar with all the methods used in this paper but they seem appropriately referenced.  I’ve pointed out where more information is needed.  The modeling of the tower with a mesh model is very innovative and worked out well.  With slight revision I feel this paper is appropriate for publication.

Line 87:  Sentence needs a verb.  Should be “was generated”.

Line 114:  What type of geophone?  What frequency range was it?

Line 160:  What are H2/V2 and H2/H1?  There needs to be a line of explanation here.  What about H3/V3?

Line 199:  define event kurtosis.  And what is “the 5-Hz frequency class”?

Line 210:  There must be a standard reference for k-means cluster analysis that can be put here.

Paragraph line 383.  Figure 10.  There are only three classification parameters shown in figure 10.  Is this all that were used?  Exactly what were they?

Station S3 did not have the same complete analysis done on it as S1.  (eg. Where is fig 11 and 12 for s3?)  Why was S3 not looked at?

Figure 11:  What are the arrows?  I have a hard time deciphering them from the text but think they are precipitation.  They should be identified as such in the caption.

Why is f2>f1?  I think it’s because the column is thicker in that direction.  Is that correct?

Author Response

We thank Reviewer 1 for the positive feedback on the manuscript. We have implemented all the comments and suggestions in the new version of the manuscript. Detailed answers to the comments are provided in the attachment.

Reviewer 2 Report

Referee’s comments on "Ambient seismic noise and microseismicity monitoring of a prone-to-fall quartzite tower (Ormea, NW Italy) " (remotesensing-1186345)

In this study, a six-month seismic monitoring was conducted on a quartzite tower to investigate and continuously monitor site stability. Ground motion directions were retrieved to understand the fracture control on tower vibrations, supported and verified by 3D numerical modeling. It is a topic of interest to the researchers in the related areas. In my point of view, the paper includes interesting research and it is worth publishing. However, several clarifications on their procedure are required in the review as follows:

  1. Some of the keywords are too long. Please make it shorter. For example, passive seismic monitoring of landslides. It can be divided into two words, i.e., landslides, passive seismic monitoring.
  2. Lines 30-31: Continuous passive seismic monitoring has now reached a decade of applications on gravitational movements of all types and geometries. I think it is more than one decade. Please check the duration.
  3. What is seismic monitoring? It should be given in the text. Several references were given as follow as example. For example, the MS monitoring technique involves using a spatially distributed set of MS sensors with different azimuths which subsequently capture the MS waves released when the rocks fracture. By analyzing the detected MS waves, the time, location, intensity, and type of rock fracture occurring can be deduced.

[1].Characteristic microseismicity during the development process of intermittent rockburst in a deep railway tunnel. International Journal of Rock Mechanics and Mining Sciences.2019, 124, 104135.

Meanwhile, if possible, the sensors should better be installed in drilled boreholes into the rockmass to optimize the network geometry. Examples can also be found in the references given.

  1. In lines 462-463: However, event source location is limited by the network geometry and do not allow for confirming this hypothesis and further locating the two clusters. There are four sensors in the test site and monitoring network. Some research can be done on the event source location in the further.

Author Response

We thank Reviewer 2 for the positive feedback on the manuscript. We have implemented all the comments and suggestions in the new version of the manuscript. Detailed answers to the comments are provided in the attachment.
